# Fucoxanthin Alleviates Oxidative Stress through Akt/Sirt1/FoxO3α Signaling to Inhibit HG-Induced Renal Fibrosis in GMCs

**DOI:** 10.3390/md17120702

**Published:** 2019-12-12

**Authors:** Guanyu Yang, Lin Jin, Dongxiao Zheng, Xiaoliang Tang, Junwei Yang, Lingxuan Fan, Xi Xie

**Affiliations:** 1Key Laboratory of Tropical Biological Resources of Ministry of Education, Hainan University, Haikou 570228, China; yangguanyu@hainanu.edu.cn (G.Y.);; 2School of Life and Pharmaceutical Sciences, Hainan University, Haikou 570228, China

**Keywords:** diabetic nephropathy, fucoxanthin, FoxO3α, Akt, Sirt1, oxidative stress, fibrosis

## Abstract

As one of the main marine carotenoids, fucoxanthin has strong antioxidant activity. FoxO3α, a member of the forkhead box O family of transcription factors, plays an important role in DN by regulating oxidative stress. The activity of FoxO3α is related to its phosphorylation and acetylation status, regulated by Akt and Sirt1, a lysine deacetylase. Our study aimed to investigate whether fucoxanthin could alleviate oxidative stress and fibrosis via FoxO3α in DN and whether Akt and Sirt1 were involved. We found that in GMCs cultured in HG, fucoxanthin treatment significantly reduced the expression of FN and collagen IV, as well as reactive oxygen species generation, suggesting that fucoxanthin is beneficial to alleviate both fibrosis and oxidative stress in DN. In addition, we found that fucoxanthin decreased the phosphorylation and acetylation level of FoxO3α, reversed the protein level of FoxO3α inhibited by HG, and then promoted the nuclear transport of FoxO3α. Besides, fucoxanthin promoted the expression of manganese superoxide dismutase, a downstream target of FoxO3α. Furthermore, we found that fucoxanthin reversed the activation of Akt and inhibition of Sirt1. However, the enhancement of fucoxanthin in FoxO3α expression and nuclear transport was significantly decreased by pretreatment with Akt activator SC79 or Sirt1 inhibitor EX527. In summary, our study explored fucoxanthin alleviated oxidative stress and fibrosis induced by HG through Akt/Sirt1/FoxO3α signaling in GMCs, suggesting fucoxanthin is a potential therapeutic strategy for DN.

## 1. Introduction

Diabetic nephropathy (DN), a serious and highly dreaded microvascular complication of diabetes mellitus (DM), has become the leading cause of end-stage renal disease. The early DN is characterized histologically by glomerular hypertrophy, glomerular basement membrane (GBM) thickening and accumulation of extracellular matrix (ECM) components and glomerulosclerosis [1]. Glomerular mesangial cells (GMCs) stimulated by hyperglycemia, which generate ECM, such as fibronectin (FN) and collagens, promote the pathological processes of glomerular matrix overproduction and glomerulosclerosis, finally leading to DN [2]. So far, the detailed mechanism of DN is not fully explored. It is believed hyperglycemia-triggered oxidative stress is mainly responsible for the progression of DN.

FoxO3α is a member of the forkhead box O family of transcription factors, which also includes FoxO1, FoxO4, and FoxO6. As transcription factors, FoxOs regulate multiple gene expression and cellular functions, particularly those related to stress response, cell growth, cell survival, and longevity [3]. FoxOs play an important role in maintaining intracellular ROS homeostasis. FoxO3α has been shown to induce several genes that protect against ROS suggesting that they play a critical role in keeping cellular ROS low by regulating oxidative defenses such as manganese superoxide dismutase (MnSOD) and catalase (CAT) genes expression [4]. It has been reported that ablation of the three FoxO genes (1, 3, and 4) in mature β-cells results in early-onset, maturity-onset diabetes of the young (MODY)-like diabetes, indicating the importance of FoxOs in insulin secretion [5]. Increasing phosphorylation of FoxO3α was found in rat and mouse renal cortical tissues two weeks after the induction of diabetes by streptozotocin [6]. However, the specific role of FoxO3α in the pathological process of DN is not yet clarified.

Silent information regulator T1 (Sirt1) is the nicotinamide adenine dinucleotide (NAD^+^) dependent protein lysine deacetylase of the sirtuin family with many physiological functions such as regulation of energy, inflammation, neuronal signaling, cell survival, DNA repair, tissue regeneration, and stress responses. Sirt1 is widely expressed in the human body, including the cells of the kidney, such as podocyte, tubular cells and GMCs. Through deacetylating substrates, Sirt1 participates in the regulation of autophagy, energetic homeostasis, mitochondrial biogenesis, and apoptosis. In recent years, researchers also have focused on the role of Sirt1 in the development and progression of DN. In kidneys, several studies have shown a critical role of Sirt1 in protecting tubular cells from cellular stresses [7,8]. A decrease of Sirt1 expression was found in DN and knockdown of Sirt1 expression could abolish the protective effect of an active component against renal damage in DN [9], while activating Sirt1 reversed the inflammation and fibrosis induced by high glucose (HG) in rat GMCs [10]. It has been suggested Sirt1 is a critical regulator of FoxO3α-mediated transcription in response to oxidative stress and cell survival by regulating FoxO3α deacetylation [11,12]. Resveratrol (Rsv), an activator of Sirt1, ameliorated renal tubular damage in oxidative, stress-related DN by upregulation FoxO3α transcriptional activity. However, the studies of Sirt1/FoxO3α signaling in the fibrosis of DN are relatively limited.

Serine-threonine kinase (Akt), which is also known as PKB is closely associated with protein synthesis and cellular growth and hypertrophy [13]. Numerous studies have reported the important role of Akt in the pathologic process of DN. Activated Akt/mTORC signaling has been observed in glomeruli in diabetes-induced by STZ injection [14]. Inhibiting Akt activity showed a protective effect for renal fibrosis and early glomerular pathological changes in DN [15]. Akt may directly phosphorylate FoxO1, FoxO3, and FoxO4 at three conserved sites, resulting in nuclear export and the consequent functional inhibition [16,17]. The overproduction of ROS promotes the nuclear localization of FoxOs and its subsequent transcriptional activities [18]. In contrast, depending on the cellular context, oxidative stress enhances Akt activity, thereby inactivating FoxO3α by phosphorylation of serine 253 in FoxO3α [19]. Thus, it seems that the distinct effects of Akt play an important role in determining the functional activity of FoxO3α.

Fucoxanthin (Fx), a xanthophyll derivative, is an orange pigment present in brown algae, which has a unique structure that includes an allenic bond, a conjugated carbonyl, a 5,6-monoepoxide, and also acetyl groups [20]. Moreover, it is metabolized to fucoxanthinol and amarouciaxanthin in vivo [21]. The therapeutic properties of Fx have been widely reported, such as anti-oxidant, anti-cancer, anti-inflammatory, anti-angiogenic, anti-obese and anti-diabetic activities [22]. Fx could improve insulin signaling and reduce hyperglycemia-associated complications in diabetic patients by inhibiting advanced glycation end-product formation. Fx has also been shown to normalize hyperglycemia and hyperinsulinemia in diabetic mice [23]. However, whether the Fx possess renoprotective effects in DN and the potential mechanisms remain to be elucidated. Hence, our study aimed to investigate whether Fx could alleviate oxidative stress and fibrosis via FoxO3α in DN and whether Akt and Sirt1 were involved in the procedure.

## 2. Results

### 2.1. Fx Could Effectively Reverse the Increase of Extracellular Matrix Induced by HG in Gmcs

FN and collagen IV are important components of mesangial ECM. The accumulation of mesangial ECM is one of the typical characteristics of DN [24]. Here, we found that Fx can effectively reduce the expression of FN and collagen IV induced by HG in GMCs, suggesting that Fx can alleviate the accumulation of ECM in DN (Figure 1a,b). Rsv was used here as an activator of Sirt1, and MK2206 as an inhibitor of Akt, both of which have the effect of reversing the elevated expression of ECM induced by HG. The results suggest that Fx may have a similar effect of activating Sirt1 meanwhile inhibiting Akt, but its specific mechanism and reliability need to be further explored. In addition, the fluorescence of FN in the HG group was strong and bright, indicating that the expression of FN increased. The fluorescence of FN in Fx, Rsv, and MK2206 treatment groups was dark and weak, indicating that expression of FN induced by HG was inhibited, which was consistent with the results of western blot (Figure 1c).

### 2.2. Fx Could Effectively Reverse the Activation of Akt and the Inhibition of Sirt1 Induced by HG in Gmcs

We next explored the changes of Akt and Sirt1 in HG-induced GMCs and pharmacological effects of Fx on them. First, we found that Akt was activated in a time-dependent manner with HG stimulation (Figure 2a). Fx and MK2206, a specific inhibitor of Akt, successfully reversed the activation of Akt induced by HG (Figure 2b). Sirt1 plays an important role in DN. Overexpression of Sirt1 has the effect of anti-oxidative stress and reducing inflammation. Here, we observed the expression of Sirt1 was inhibited by HG. The Sirt1 activator Rsv effectively promoted the protein level of Sirt1 under HG, and Fx produced similar pharmacological effects to Rsv (Figure 2c). In conclusion, Fx effectively reversed the activation of Akt and the inhibition of Sirt1 induced by HG in GMCs.

### 2.3. Fx Reverses the Expression of Foxo3α Inhibited by HG in Gmcs

Previous studies have shown that the activation of FoxO3α has the effect of anti-oxidative stress [25]. Since the activation of Akt and the inhibition of Sirt1 expression induced by HG can be effectively reversed by Fx, we further explored the expression of FoxO3α under HG stimulation and the regulation of Fx on it. We designed a high-glucose stimulation experiment in a time-dependent manner. The results showed that the expression of FoxO3α was the lowest at 12 h of HG treatment, while phosphorylated FoxO3α (S253) was the highest at this time point (Figure 3a). Previous studies have shown that phosphorylation at this site will result in cytoplasmic transform of FoxO3α and decrease its transcriptional activity [26]. It is gratifying to see that Fx can successfully reverse the decrease expression of FoxO3α and the increase of phosphorylated FoxO3α induced by HG. Rsv and MK2206, activators of Sirt1 and inhibitors of Akt, respectively, were used as positive control, which also successfully increased the expression of FoxO3α and decreased the phosphorylation of FoxO3α in GMCs stimulated by HG (Figure 3b,d) The acetylation modification of FoxO3α, similar to phosphorylation modification, is not conducive to its transcriptional activity [27]. In this experiment, we found that the expression of acetylated FoxO3α in GMCs stimulated by HG increased significantly; however, Fx and Rsv decreased the acetylated FoxO3α induced by HG, respectively (Figure 3c).

### 2.4. Fx Regulates the Expression of Foxo3α through the Akt and Sirt1 Signaling in Gmcs

It has been reported that both Akt and Sirt1 can regulate FoxO3α [27]. To further determine whether Fx regulates FoxO3α through Akt and Sirt1 signaling, we treated GMCs with EX527, an inhibitor of Sirt1, and SC79, an activator of Akt, followed by Fx, respectively. The results showed that Fx restored the expression of FoxO3α inhibited by HG but could not restore the expression of FoxO3α in GMCs pretreated with EX527 and SC79, respectively (Figure 4a). In addition, Fx inhibited phosphorylation and acetylation of FoxO3α induced by HG successfully but could not inhibit the acetylation and phosphorylation of FoxO3α in GMCs pretreated with EX527 and SC79, respectively (Figure 4b,c). These results suggest that the regulation of Fx on FoxO3α depends on Akt and Sirt1 signaling.

### 2.5. Fx Enhances the Nuclear Transport of Foxo3α through Akt and Sirt1 Signaling in Gmcs

To confirm whether Fx could enhance nuclear transportation of FoxO3α through Sirt1 regulation, we firstly detected the expression of FoxO3α in the nucleus of GMCs. The results showed that HG treatment decreased the FoxO3α in the nucleus; however, increased FoxO3α was observed in the nucleus after Fx treatment (Figure 5a). It suggests that Fx can effectively promote the nucleus transport of FoxO3α inhibited by HG. However, the addition of Sirt1 inhibitor EX527 abolished Fx-induced FoxO3α nuclear transportation. It suggests that the regulation of Fx on the nucleus transport of FoxO3α depends on Sirt1. Akt activation promotes the phosphorylation of FoxO3α, thus enhancing the cytoplasmic transform of FoxO3α, and reduces its transcriptional activity [28]. It was observed here Akt inhibitor MK2206 significantly increased the FoxO3α in the nucleus, while Akt activator SC79 abolished concentration of FoxO3α in the nucleus induced by Fx (Figure 5b). It suggests the important role of Akt signaling in the regulation of Fx on FoxO3α. In addition, we confirmed this conclusion from the immunofluorescence results of FoxO3α in GMCs. In the HG group, weak fluorescence of FoxO3α was observed, while it increased significantly after treatment of Fx, Sirt1 activator, and Akt inhibitor, respectively (Figure 5c). These results suggest the regulation of Fx on the nucleus transport of FoxO3α depends on Sirt1 and Akt signaling in HG-induced GMCs.

### 2.6. Fx Effectively Increases the Expression of Mnsod, Reduces the Oxidative Stress Level, and Improves Mitochondrial Membrane Potential in HG Induced Gmcs

It has been reported that Fx increases antioxidant enzyme activity and mitochondrial biogenesis, reduces superoxide production and oxidative stress [29]. Oxidative stress plays an important role in causing tissue damage, stimulating the accumulation of inflammatory factors, and aggravating the process of DN [30]. Thus, we further confirmed whether Fx decreased oxidative stress through Akt/Sirt1/FoxO3α signaling. The western blot and immunofluorescence results of MnSOD (Figure 6a,b) showed that HG inhibited the expression of MnSOD, while Rsv or MK2206 could partially restore the expression of MnSOD inhibited by HG. Fx was more effective than both in promoting the expression of MnSOD in the HG condition (Figure 6a) However, inhibition of Sirt1 or activation of Akt significantly reduced the promoting effect of Fx on the expression of MnSOD (Figure 6b), suggesting that Fx regulates the expression of MnSOD by activating the transcriptional activity of FoxO3α through Akt and Sirt1 signaling. We also detected the production of ROS in GMCs stimulated by HG. The results showed that HG stimulated the overproduction of ROS in GMCs, indicating oxidative stress occurred. Fx effectively inhibited the overproduction of ROS induced by HG in an Akt/Sirt1/FoxO3α-dependent manner (Figure 6c). Furthermore, we explored the changes of the mitochondrial membrane potential of GMCs under HG condition. The results showed that the membrane potential decreased significantly in the HG group, which was counteracted by Fx (Figure 6d). These results suggest Fx reduces the oxidative stress level and maintains the mitochondrial membrane potential of GMCs cultured in HG by activating regulating Akt/Sirt1/FoxO3α signaling, which further reduces the cell damage and apoptosis induced by oxidative stress.

## 3. Discussion

Although relatively fewer studies of Fx on DN, several studies have reported bioactivities of Fx about inflammation inhibition and antioxidant properties, indicating its potential anti-diabetic role. In rat models, pretreatment with Fx inhibited UV-B radiation induced corneal disorders including evident preservation of corneal surface smoothness, downregulation of proinflammatory cytokine expression, and decrease of infiltrated polymorphonuclear leukocytes from UV-B induced damage [31]. The combination of Fx and rosmarinic acid improved antioxidant and anti-inflammatory profiles through the down-regulation of inflammasome components, such as NLRP3 and ASC, and increased Nrf2 and HO-1 antioxidant genes expression in HaCaT Keratinocytes [32]. Fx protected UV-B induced cell damage by decreasing intracellular ROS generation [33]. In addition, Fx exhibited a big therapeutic index according to the report that a single oral dose of 2000 mg/kg and repeated oral dose of 1000 mg/kg in the toxicity study of Fx revealed no mortality and no abnormalities in mice [34]. As such, in our experiment, Fx administration significantly decreased ECM secretion and ROS generation. However, the specific mechanisms responsible for the renal protective effect of Fx in DN remain unclear.

DN was a progressive microvascular complication primed by hyperglycemia through an integrated signaling network of metabolism disorder, chronic inflammation and oxidative stress, and was the leading cause of renal failure [35]. Many pathological factors merged in DN and make it a complication difficult to cure. Among all factors, oxidative stress plays an important role in DN. Upregulation of inflammatory through NF-κB signaling and ROS generation through HG-inhibited Nrf2/ARE signaling promoted DN progression [36,37]. In the present study, increased ROS generation and ECM secretion induced by HG were both decreased after Fx treatment. Rsv possessed renoprotective nature by attenuating oxidative stress in renal tissues of diabetic rats [38]. However, Fx at the concentration of 2 μM was almost equal to 40 μM Rsv treatment in the alleviation expression of FN and collagen IV, which revealed the potent renoprotective effect of Fx. These finding suggest Fx can interfere with the pathogenesis of DN by reducing the accumulation of ECM components and ROS generation.

As a nuclear transcription factor mediate oxidative stress, FoxO3α participates in multiple signaling. It has been reported that FoxO3α regulates the cell resistance of oxidative stress by encoding the oxidative stress resistance gene [39]. Overexpression of FoxO3α results in an increase in both hydrogen peroxide scavenging and oxidative stress resistance [13]. Decreased FoxO3α expression has been observed in our HG cultured GMCs, which may explain for the increased ROS generation and oxidative stress damage. The transcriptional activity of FoxO3α is regulated by several post-translational mechanisms, including phosphorylation, acetylation, and ubiquitination. Based on a unique structural feature in the C-terminal region, FoxO3α-DNA binding domain C-terminal phosphorylation by Akt or acetylation by cAMP-response element-binding protein (CBP) can attenuate the DNA-binding activity and thereby reduce transcriptional activity of FoxO3α [40]. In our studies, Sirt1 as a NAD^+^ dependent deacetylase, its activity was inhibited by high glucose (HG), which results in a higher degree of acetylation of FoxO3α in HG group. The activity of Sirt1 was restored or partially restored in the administration group (Fx and resveratrol), which results in the decrease of FoxO3α acetylation degree. Accumulated nuclear p-Akt leads to the inactivation of FoxO3α-mediated transcription of pro-apoptotic Bim and the cell cycle inhibitor p27 (kip1) [41]. PTEN, an Akt inhibitor, increased nuclear localization of the transcription factor FoxO3α [42]. It has been reported that activated Akt inhibits FoxO3α, causing decreased apoptosis and promoted proliferation. Entirely activated Akt phosphorylated FoxO3α in the NLS sequence [35], thus enhanced its extra-nuclear transportation and decreased FoxO3α transcription activity. However, whether activated Akt signaling negatively regulated the antioxidative effect of FoxO3α in DN remained unclear. In our studies, Akt was activated by HG, which was related to decrease FoxO3α and increased ROS generation. However, treatment with Fx or MK2206, an Akt inhibitor, promotes FoxO3α expression and decreases ROS production. HG-induced activation of Akt negatively regulated FoxO3α in antioxidant stress and Fx reversed it by inhibiting Akt activation.

As an upstream mediator of FoxO3α, Sirt1 plays an important role in DN progression. Sirt1 activation alleviates DN progression and enhances antioxidant ability. Rsv or Sirt1 overexpression markedly increased Sirt1 levels and reduced FN and TGF-β1 expression [43]. Sirt1 deacetylates FoxO3α in response to oxidative stress hence increases FoxO3α DNA binding and elevates the expression of FoxO3α target genes, such as p27(Kip1) and MnSOD [44]. Sirt1 results in the deacetylation and modulation of the activity of downstream targets that include the peroxisome proliferator-activated receptor-gamma coactivator 1α, FoxO1, and FoxO3α [45]. In our studies, Sirt1 suppression by HG always correlated with a low level of FoxO3α expression and decreased mitochondria MnSOD expression and ascendant ROS level. However, overexpression of Sirt1 through Rsv or Fx treatment restored FoxO3α level, increased MnSOD expression, and downregulated ROS generation. These provided the specific mechanism of Fx to resistant oxidative stress through inducing Sirt1 expression, which deacetylates acetylated FoxO3α and increases their transcription activity, thus increasing MnSOD expression and decreasing ROS generation.

Mitochondria play a crucial role in diabetic kidney disease. Mitochondrial production of ROS in response to chronic hyperglycemia may be the key initiator for pathogenic pathways that inducing DN [46]. This emphasizes the importance of mitochondrial dysfunction in the progression and development of diabetes complications including DN. In the mitochondria, the unbalance of increased ROS generation and decreased antioxidant enzyme lead to oxidative stress, which accelerated DN progression. There has been reported that mitochondrial abnormalities, such as defective mitophagy, ROS overexpression, and mitochondrial fragmentation, occurred in the tubular cells of *db/db* mice, and in HG-induced HK-2 cells [47]. In our HG cultured GMCs, mitochondrial membrane potential declined, and ROS increased, which accompanied lower expression of MnSOD. However, these effects were reversed by improved FoxO3α level through Fx treatment by inhibition of Akt and activation of Sirt1.

In the current study, we found that Fx treatment alleviated ROS generation and ECM accumulation in HG-induced GMCs. Moreover, HG-induced the inhibition of Sirt1 and activation of Akt, which both enhanced FoxO3α nuclear exportation and decreased transcriptional activity. However, Fx treatment strongly promoted the nuclear translocation and transcriptional activity of FoxO3α as well as upregulated the expression of MnSOD, ultimately quenching the higher level of ROS and inhibiting the FN and collagen expression induced by HG in GMCs.

## 4. Materials and Methods

### 4.1. Reagents and Materials

NG dulbecco’s modified eagle medium (DMEM) and HG DMEM were purchased from Gibco (New York, NY, USA). Fetal bovine serum (FBS) of Si Ji Qing and buffer solution like TBS and PBS were from Lianshimall (Beijing, China). Fx (98%) was purchased from Biopurify (Chengdu, China). Akt inhibitor MK2206, Akt activator SC79, Rsv, and Sirt1 inhibitor EX527 were purchased from Beyotime (Haimen, China). ROS fluorescence probe, RIPA Lysis Buffer, Cytoplasm and Nuclear Protein Extraction Kit, beyoECL Plus, 30% Acr-Bis, and BCA Protein Assay Kit were purchased from Beyotime (Haimen, China). Antibody against FN (ab199056) was obtained from Abcam (Cambridge, UK). Antibodies against collagen IV (BA3585-2), p-FoxO3α (S253, BM4401), Akt1/2/3 (BM4400), p-Akt (S473, BM4838) were purchased from Boster (Wuhan, China), and antibody against FoxO3α (66428), pan acetylated antibody (66289), MnSOD (24127) were from Proteintech (Wuhan, China). Antibody against Sirt1 (K106486P) was from SolarBio (Beijing, China). Antibodies against tubulin (AT819-1), actin (AA128-1), histone H3 (AH433) were purchased from Beyotime (Haimen, China), second antibodies HRP conjugated goat anti-rabbit (BA1054), HRP conjugated goat anti-mouse (BA1050), dylight488- labeled goat anti-mouse (BA1126), dylight550-Labeled goat anti-rabbit (BA1135) were purchased from Boster (Wuhan, China).

### 4.2. Cell Culture

GMCs were extracted from the cortex of young Sprague–Dawley (SD) rat kidney and identified as previously reported [48]. Cells were cultured in NG DMED medium with 10% FBS addition, and cells within 10 generations were used for experiments. Confluent cells were treated with serum-free medium for 24 h to keep them at the quiescent stage [36]. The cells then were treated with 30 mM HG medium or 5.6 mM NG medium mixed with drugs for indicated time.

### 4.3. Western Blot

After treatment, cells were lysed and extracted total protein with RIPA lysis buffer pre-mixed with 1mM PMSF or extracted the nuclear and cytoplasmic protein by cytoplasmic and nuclear protein extraction kit per manufacturer’s instruction. In addition, phosphatase inhibitors were needed for the determination of p-Akt (S473) and p-FoxO3α (S253). After high-speed freezing centrifugation, lysis supernatant was absorbed for subsequent experiments and avoided touching precipitation. The protein supernatant was quantified by BCA Assay Kit and next packed into EP tube in a dose of 30 μg. The packed protein was denatured in boiling water for 5 min after mixed with DTT and loading buffer. The denatured protein was cooled down at room temperature and then instantaneously centrifuged to keep mixture at the bottom of the EP tube. The protein was separated by 10% polyacrylamide gel electrophoresis and then transferred to nitrocellulose membrane. After that, the membrane was washed per 10 min for three times with TBST and next blocked with 5% skimmed milk at room temperature for 1 h, after that membrane was incubated with target first antibodies at 4 °C overnight. The next day, the membrane was washed per 10 min for three times, then incubated with the second antibody diluent at room temperature for 1 h. When incubation finished, the membrane was washed again as previous. Finally, the membrane was incubated with beyoECL Plus for 1–2 min in dark condition and the signals were captured by a chemiluminescent device (JUNYI Capture, Beijing Jun Yi, China). The gray value of the protein bind was analyzed by Gelpro32 software.

### 4.4. Immunofluorescence

After treatment, the culture medium was removed, and cells were washed with PBS for three times. Next, Cells were fixed with 4% paraformaldehyde at room temperature for 15 min and washed with PBS for three times. After that, fixed cells were permeated with 1% Triton X-100 for 20 min at room temperature and then washed again as previously. Permeated cells were blocked with goat serum at room temperature for 30 min. When finished, the goat serum was sucked out and leave the slide obliquely to make water drain away. Then, circles were drawn with hydrophobic barrier pen, and the first antibody diluent was added to the sliding circle and incubated overnight at 4 ℃. The next day, the slides were washed with TBST per 5 min for three times to remove the free first antibody. Then, the cover slides were incubated with fluorescence-labeled second antibody for 1 h in dark condition at room temperature. The slides were washed three times as previous before and after incubated with DAPI nucleus staining solution. Finally, slide specimens were made by covering the cell-contained slide into another slide with fluorescent mounting media and fixed slides by surrounding the slide with nail polish. All the processes including washing, sealing and photographing were performed under dark conditions after the second antibody incubation. The prepared slide was photographed under an Olympus IX71 fluorescence microscope (Olympus Inc. Japan) using Image-Pro Plus6.0 software.

### 4.5. Determination of ROS

GMCs were treated with HG to induce the ROS production, and Rosup (10 μM) (Beyotime, China) was treated for 30 min as the positive control of ROS production. The staining processes were carried out in a dark environment with reactive oxygen species assay kit (Beyotime, Haimen China). DCFH-DA was prepared with a serum-free medium at a concentration of 10 μM and incubated with the serum-free cells for 20 min in a 37 ℃ incubator. Once incubation finished, cells were washed with PBS for three times to remove out excess probes to reduce the fluorescence background and then incubated with Hoechst3342 nuclear staining solution at a concentration of 20 μM in 37 ℃ incubator for 10 min, and the excess Hoechst 3342 was washed out with PBS for three times after that. Finally, cells were contained in a serum-containing medium and photographed under an Olympus IX71 fluorescence microscope, with emission wavelength of 525 nm and excitation wavelength of 488nm for ROS and emission wavelength of 460 nm and excitation wavelength of 346 nm for Hoechst3342.

### 4.6. Statistical Analysis

All the experiments were repeated at least three times with similar results. GelPro32 (Media Cybernetics, Silver Spring, MD, USA) was used to quantify the gray value of Western blot bind. SPSS Statistics software was used for statistical analysis with one-way ANOVA. All values are expressed as mean ± SD. *p* < 0.05 was considered to be statistically significant.

## 5. Conclusions

In conclusion, our study revealed that Fx reduced the phosphorylation and acetylation of FoxO3α through Akt/Sirt1 signaling, promoted the nuclear transport of FoxO3α, further increased the expression of MnSOD, reduced the production of ROS, and alleviated oxidative stress induced by HG in GMCs. Our study demonstrated a new mechanism of Fx in the treatment of DN.

## Figures and Tables

**Figure 1 marinedrugs-17-00702-f001:**
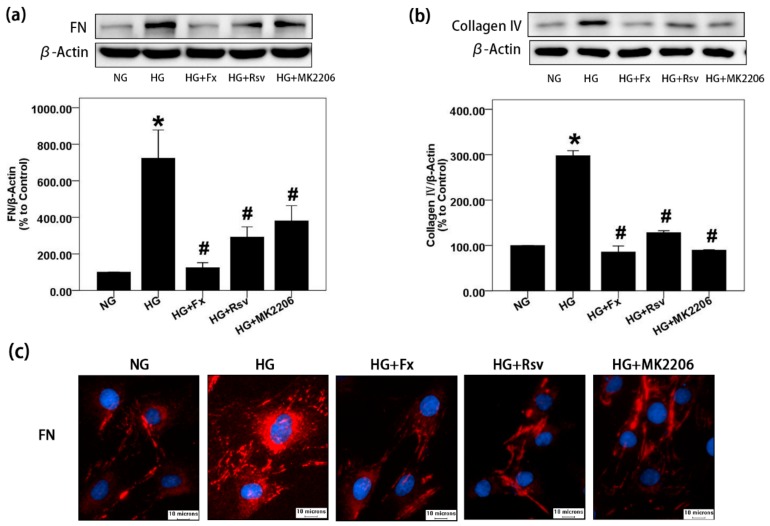
Fx can significantly reduce the accumulation of FN and collagen IV induced by HG in GMCs. (**a**) FN was detected by western blot after 24 h of HG treatment with or without Fx (2 μM), Rsv (40 μM) or MK2206 (2 μM). (**b**) After 24 h of HG treatment with or without Fx (2 μM), Rsv (40 μM) or MK2206 (2 μM), GMCs were lysed for protein extraction, and collagen IV was detected by western blot. (**c**) GMCs were induced by HG with or without Fx (2 μM), Rsv (40 μM), or MK2206 (2 μM) for 24 h. FN was stained by the immunofluorescence with antibody against FN and then captured under Olympus IX71 (400X); red fluorescence indicates FN, blue fluorescence indicates nucleus, and scale represents 10 μm. β-Actin was measured as a loading control; all the experiments were repeated at least three times independently with similar results. * *p* < 0.05 vs. normal glucose (NG) group and # *p* < 0.05 vs. 30 mM HG group.

**Figure 2 marinedrugs-17-00702-f002:**
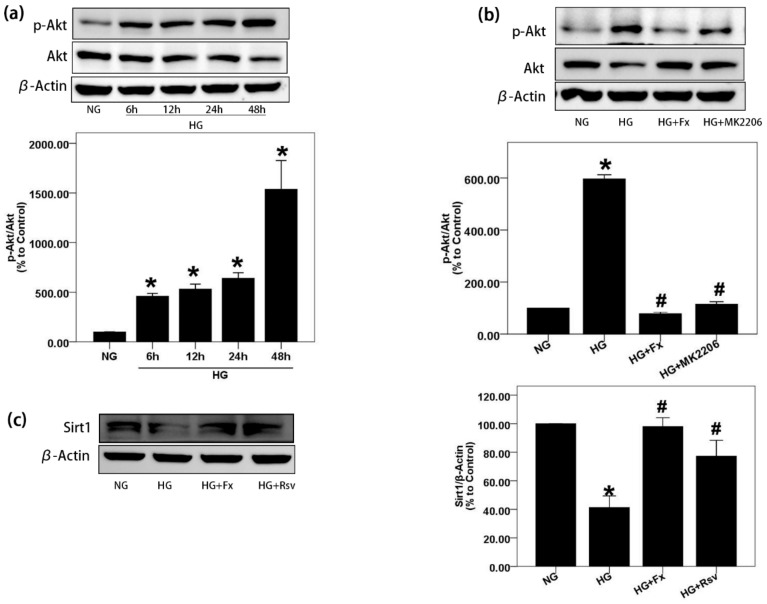
Fx reverses the activation of Akt and the inhibition of Sirt1 induced by HG. (**a**) GMCs were treated with HG for indicated times and protein was extracted for p-Akt (S473) detection by western blot. (**b**) GMCs were treated with HG with or without Fx (2 μM) or MK2206 (2 μM) respectively for 24 h, and protein was extracted for p-Akt (S473) and Akt detection by western blot. (**c**) After treated with HG with or without Fx (2 μM) or Rsv (40 μM) respectively for 24 h, GMCs were lysed and protein was extracted for Sirt1 quantification by western blot. β-Actin were measured as the loading control; experiments were repeated at least three times with similar results. * *p* < 0.05 vs. NG group and # *p* < 0.05 vs. HG group.

**Figure 3 marinedrugs-17-00702-f003:**
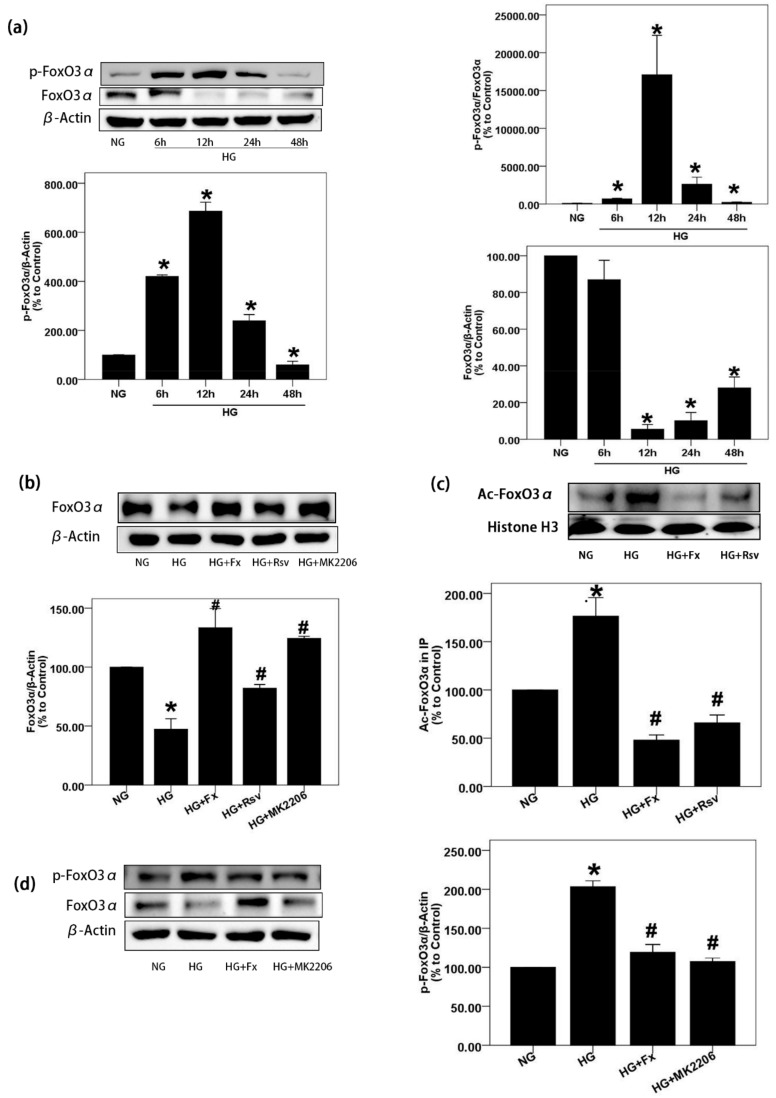
Fx reverses the lower expression of FoxO3α in GMCs induced by HG. (**a**) GMCs were treated by HG for indicated times, and cells were lysed to extract protein for the detection of FoxO3α and p-FoxO3α (S253) by western blot. (**b**,**d**) GMCs were treated by HG combined with or without Fx (2 μM), Rsv (40 μM), or MK2206 (2 μM) for 24 h, then protein was extracted for the detection of FoxO3α and p-FoxO3α (S253) by western blot. (**c**) GMCs were treated as previous and nuclear protein was obtained. Total FoxO3α was precipitated by incubating with antibody against FoxO3α over the night under 4 °C, and acetylated FoxO3α were displayed by pan acetylation antibody. Histone H3 and β-actin were measured as the loading control, experiments were repeated at least three times with similar results. * *p* < 0.05 vs NG group and # *p* < 0.05 vs 30 mM HG group.

**Figure 4 marinedrugs-17-00702-f004:**
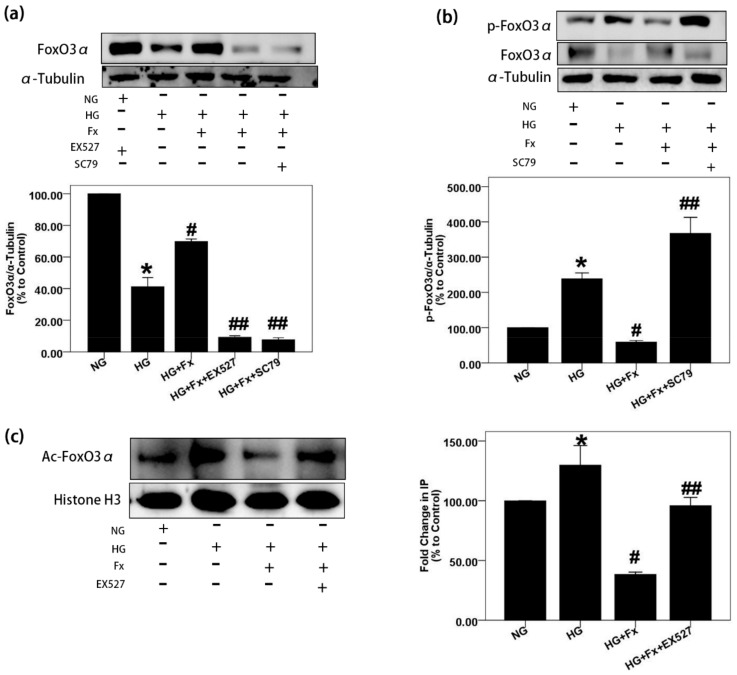
Fx regulates FoxO3α through Akt and Sirt1 signaling in GMCs. (**a**,**b**) GMCs were treated with NG (5.6 mM), HG (30 mM), HG combined with Fx (2 μM), HG combined with Fx (2 μM) and EX527 (1 μM), or HG combined with Fx (2 μM) and SC79 (20 μM), respectively, for 24 h. Protein was obtained, and western blot was performed to detect FoxO3α and p-FoxO3α (S253). (**c**) GMCs were treated as previously, and nuclear protein was extracted for immunoprecipitation. The lysis supernatant was quantified by BCA assay and packed equally. Total FoxO3α was precipitated by incubating with an equal FoxO3α antibody and performed the resuspension and centrifugation processes for five times after that. Mixture of protein and loading buffer were boiled for 5 min, and then, a process similar to western blot experiment was performed for the acetylated FoxO3α detection with pan acetylation antibody. α-Tubulin and histone H3 were measured as the loading control; experiments were repeated three times with similar results. * *p* < 0.05 vs. NG group, # *p* < 0.05 vs. 30 mM HG group, ## *p* < 0.05 vs. HG+Fx group.

**Figure 5 marinedrugs-17-00702-f005:**
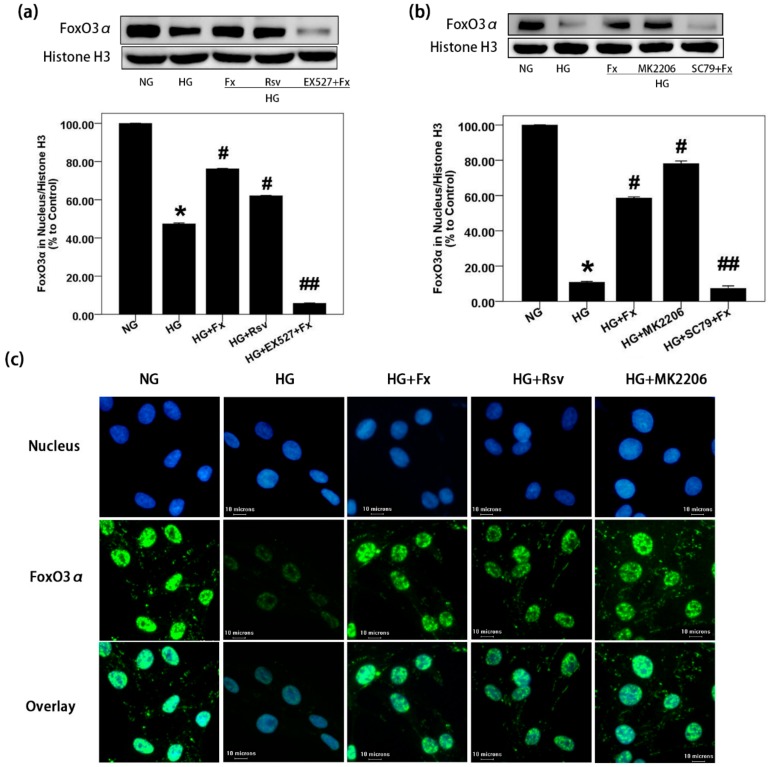
Fx enhances nuclear transport of FoxO3α in HG-induced GMCs, depending on Sirt1 and Akt signaling. (**a**,**b**) GMCs were treated with NG (5.6 mM), HG (30 mM), HG combined with Fx (2 μM), Rsv (40 μM) or MK2206 (2 μM), HG combined with Fx and EX527 (1 μM), HG combined with Fx and SC79 (20 μM), respectively, for 24 h. Cells were lysed, and nuclear protein was obtained for western blot, to detect FoxO3α in the nucleus. (**c**) Immunofluorescence assay was performed after GMCs being treated by HG with or without Fx (2 μM), MK2206 (2 μM), or Rsv (40 μM), respectively. The blue fluorescence represents the nucleus, the green fluorescence represents FoxO3α, and the scale represents 10 μm (400X). Histone H3 was measured as the loading control, experiments were repeated three times with similar results. * *p* < 0.05 vs. NG group, # *p* < 0.05 vs. HG group, ## *p* < 0.05 vs. HG+Fx group.

**Figure 6 marinedrugs-17-00702-f006:**
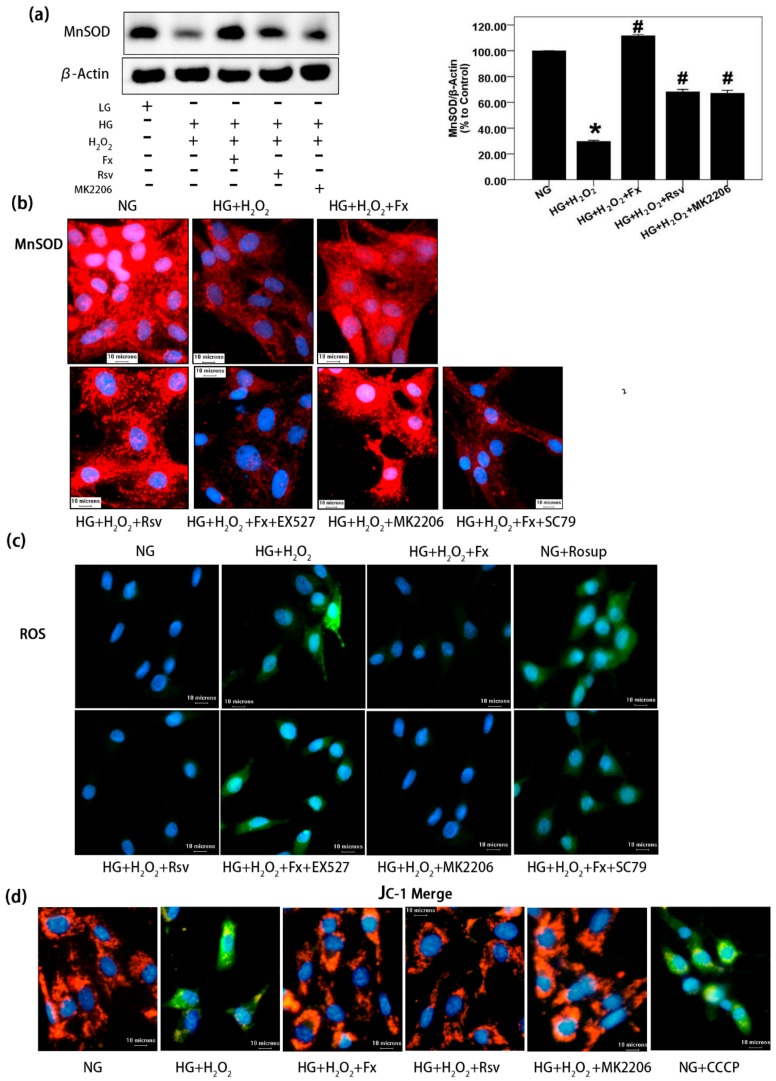
Fx alleviates oxidative stress of GMCs cultured in HG through Akt/Sirt1/FoxO3α signaling. (**a**) After 24 h serum-free medium treated, the GMCs were induced with 150 μM hydrogen peroxide-contained HG, combined with or without Fx (2 μM), Rsv (40 μM), or MK2206 (2 μM), respectively, for 24 h. Then protein was extracted, and MnSOD was detected by western blot; β-actin was measured as the loading control, and experiments were repeated three times with similar results. * *p* < 0.05 vs. NG group, and # *p* < 0.05 vs. HG group. (**b**) The GMCs were treated with 150 μM hydrogen peroxide-contained HG combined with or without Fx (2 μM), Rsv(40 μM), MK2206 (2 μM), Fx (2 μM) combined with EX527 (1 μM), or Fx (2 μM) combined with SC79(20 μM), respectively, for 24 h, and MnSOD was detected by immunofluorescence method under Olympus IX71. Red fluorescence represents MnSOD, and blue fluorescence represents the nucleus; scale bar represents 10 um (400X). (**c**) The GMCs were induced oxidative stress with 150 μM hydrogen peroxide-contained HG combined with or without drug administration as previously, for 24 h. Rosup (10 μM) was used as a positive reagent to induce ROS for 30 min. ROS was stained with DCFH-DA for 30 min under dark condition and latter captured under Olympus IX71. Green fluorescence represents ROS; blue fluorescence represents the nucleus; and the scale represents 10 μm (400X). (**d**) GMCs were treated as previously and then incubated in the incubator with JC-1 for 30 min. Mitochondrial membrane potential was captured under Olympus IX71. Blue fluorescence represents the nucleus; green fluorescence represents JC-1 monomer; red fluorescence represents JC-1 aggregate, and the scale represents 10 μm (400 X).

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
