# Peer review of "Fucoxanthin Alleviates Oxidative Stress through Akt/Sirt1/FoxO3α Signaling to Inhibit HG-Induced Renal Fibrosis in GMCs"

_marinedrugs, 2019, doi:10.3390/md17120702_

Round 1

Reviewer 1 Report

The manuscript by Yang et al reports a protective effect of a marine carotinoid fucoxanthin (FC) against a high glucose (HG)-induced fibrogenesis in vitro. The authors clearly show that HG triggers expression and accumulation of collagen and fibronectin, whereas the drug prevents these changes. Moreover, they elucidated that the effect is achieved via AKT, SirT1, and Foxo3a signaling. The study is well designed, the data are quite convincing, and all conclusions are supported by presented data.

However, there are several flaws that should be revised prior publication.

Major concerns.

Fig 3a: please provide western blot for total Akt, as well as normalization of p-Akt for total Akt. Otherwise it is not clear why panes a and b show different normalization. Fig 3a,b: total FoxO3a levels should be provided for the same experiment, for which p-FoxO3a is provided. And why both panels use different housekeeping protein? Subsection 2.6: why effect of the compound on antioxidant defense was studied (only) by quantifying SOD2? Why this mitochondrial enzyme is connected to HG pathological effects? In addition, why the authors used DCFH2DA that reacts with hydroxyl radical (a product of peroxide decomposition via Fenton reaction) and RNS mostly in the cytoplasm, whereas SOD2 produces H2O2 from superoxide anion in mitochondria. How do the results match to each other? And even in this case ROS production should be quantified. Why such experiments were not carried out for the cells treated with SC79, MK2206 and Et527? Subsection 4.5: the authors should revise the text by giving specific details (name of the dye, its concentration, time of treatment, description of washing steps, wave lengths of fluorescence excitation and emission.

Minor issues.

Line 33: Please state for the DN abbreviation once again. 3e: acetylation should be marked as Ac, not Ace (as acetyl group is written in organic chemistry). General comment: why on some panels fucoxanthin is written with a full name, whereas on others - only with an abbreviation Fx? A legend for Fig 4: please indicate again which concentration of Fx was used. Catalogue numbers for each antibody used should be provided in Experimental section.

Author Response

Dear professor,

Thank you for your review! Please see the attachment.

Best regards!

Xi Xie

School of Life Science and Pharmacy, Hainan University

Reviewer 2 Report

General comments to the article Fucoxanthin Alleviates Oxidative Stress through Akt/Sirt1/FoxO3α Signaling to Inhibit HG-Induced Renal Fibrosis in GMCs to bu published in Marine Drugs (marinedrugs-647231): This is a very interesting piece of work giving insights in the Fucoxanthin as an active protector and regulator of the processes leading to renal fibrosis via oxidative stress. In the presented article, the authors proposed the mechanism of action of Fucoxanthin by Akt / Sirt1 / FoxO3α signaling and adequately addressed and confirmed their theory by experiments.

It indicates to Fucoxanthin usage as a potential strategy for diabetic nephropathy.
I believe that it should be published after the revision process.  However, the way of data presentation raises, in my opinio , a few questions.

Title: Different abbreviations are used both in the title and in the text - Gmcs versus GMCs; Foxo3α versus FoxO3α, please check it for consistency throughout the text.

There are many abbreviations used in the text which are not sufficiently explained, therefore it is highly recommended to insert a list of abbreviations used, especially that the authors use the same abbreviations interchangeably or the same substances are abbreviated differently than in the quoted references.

Abstract: The authors have used several experimental combinations to prove that fucoxanthin has several roles in thecuration of renal fibrosis  as: i) fucoxanthin (F) reverses the increase of extracellular matrix induced by high glucose content, ii) F reverses the activation of Akt and inhibition of Sirt1, iii) F reverses expression of FoxO3α inhibited by glucose and GMCs, iv) it regulates the expression of FoxO3α through the Akt and Sirt1 signaling in GMCs and v0 enhances its nuclear and vi) reduces the oxidative stress in GMCs.
The results are very interesting and should be summarized more clearly. In the present form the content of a paper is hardly expressed in THE Abstract PART. Besides, no unexplained abbreviations should be used in the Abstract.

Figures: Please pay attention to the resolution of the bar graphs (it looks as if they were pasted from another program). Descriptions are not uniformed (Fx+ Hg versus Hg + Fucoxanthin). Please use the same abbreviations. Due to the low resolution quality it is difficult to differentiate between star (*) and hash (#) symbols.

Name of carotenoid Fucoxanthin is spelled incorrectly in all graphic images. It should be Focuxanthin instead Fucoxanthin.

General comments to presentation of the data (Figures)

Fig. 3 a. The qualitative and quantitative results of FoxO3α expression are inconsistent
Fig. 3b Qualitative and quantitative results of p-FoxO3α expression are consistent, there is also complementarity of FoxO3α and p-FoxO3α expression according to the proposed theory.  
There is also  one question concerning the usage of appropriate references. Please explain why the references have been changed (for FoxO3α it is Beta-Actin and for p-FoxO3α it is Alpha-Tubulin, it seems more logical to use the same control).

Is it possible that the authors had doubts about the results knowing the previously published paper by Dittmer, A., Dittmer, J. (2006) β‐Actin is not a reliable loading control in Western blot analysis.  Electrophoresis 27, 2844–2845. https://doi.org/10.1002/elps.200500785?

Fig. 3c and 3d - here the results are consistent and seem to be correct, the authors use the same control (Beta-Actin)!

 Fig. 3e Here I would expect, in quantitative results compared to Western Blotting, a higher expression of acetylated FoxO3α.  Please explain that phenomenon in discussion.  
Fig. 4 Here the results from Western blotting and quantitative analysis  are consistent and correct (also the same control used), also  a vs 4b are consistent according to the theory given by the authors.

Figures 5 a and 5 b. The qualitative and quantitative results of FoxO3α expression are consistent, whereas for Fig 5 c I would expect higher expression of the FoxO3α protein in the negative control (according to results in 5 a and 5 b or lower expression of the FoxO3α protein in Rsv + HG and MK2206 + HG compared to negative control (NG)

Fig. 6 a Why is  SOD2 expression result for HG + H2O2 + Resveratrol on Western blotting higher than quantitative

Materials and Methods: I have impression that many reagents (mainly buffers) are not listed in this section

References: References are not formatted according to THE MDPI style for Marine Drugs Journal. Titles - not italicized, names of Journals - italics....see guidance for authors

Author Response

(The authors gave the same response as above.)

Round 2

Reviewer 1 Report

The authors have addressed most of my concerns.